# Complete chloroplast genome sequences of Myristicaceae species with the comparative chloroplast genomics and phylogenetic relationships among them

Changli Mao, Fengliang Zhang, Xiaoqin Li, Tian Yang, Qi Zhao, Yu Wu*

Yunnan Institute of Tropical Crops, Xishuangbanna, China

* hhyyw20030105@126.com

## Abstract

### Background

Myristicaceae was widly distributed from tropical Asia to Oceania, Africa, and tropical America. There are 3 genera and 10 species of Myristicaceae present in China, mainly distributed in the south of Yunnan Province. Most research on this family focuses on fatty acids, medicine, and morphology. Based on the morphology, fatty acid chemotaxonomy, and a few of molecular data, the phylogenetic position of *Horsfieldia pandurifolia* Hu was controversial.

### Results

In this study, the chloroplast genomes of two *Knema* species, *Knema globularia* (Lam.) Warb. and *Knema cinerea* (Poir.) Warb., were characterized. Comparing the genome structure of these two species with those of other eight published species, including three *Horsfieldia* species, four *Knema* species, and one *Myristica* species, it was found that the chloroplast genomes of these species were relatively conserved, retaining the same gene order. Through sequence divergence analysis, there were 11 genes and 18 intergenic spacers were subject to positive selection, which can be used to analyze the population genetic structure of this family. Phylogenetic analysis showed that all *Knema* species were clustered in the same group and formed a sister clade with *Myristica* species support by both high maximum likelihood bootstrap values and Bayesian posterior probabilities; among *Horsfieldia* species, *Horsfieldia amygdalina* (Wall.) Warb., *Horsfieldia kingii* (Hook.f.) Warb., *Horsfieldia hainanensis* Merr. and *Horsfieldia tetratepala* C.Y.Wu. were grouped together, but *H. pandurifolia* formed a single group and formed a sister clade with genus *Myristica* and *Knema*. Through the phylogenetic analysis, we support de Wilde' view that the *H. pandurifolia* should be separated from *Horsfieldia* and placed in the genus *Endocomia*, namely *Endocomia macrocoma* subsp. *prainii* (King) W.J.de Wilde.

### Conclusion

The findings of this study provide a novel genetic resources for future research in Myristicaceae and provide a molecular evidence for the taxonomic classification of Myristicaceae.

**Data Availability Statement:** All complete chloroplast genome sequences are available from the NCBI database (https://www.ncbi.nlm.nih.gov/)

under accession numbers MK285561-MK285565, MN683753-MN683756, and MH445411.

**Funding:** This work was supported by the National Natural Science Foundation of China (No.31960289). The funders had no role in study design, data collection and analysis, decision to publish, or preparation of the manuscript.

**Competing interests:** The authors have declared that no competing interests exist.

## Introduction

The family Myristicaceae includes about 20 genera and 500 species of evergreen trees distributed from tropical Asia and Pacific islands to Africa and tropical America; there are 3 genera and 10 species of Myristicaceae in China [1]. The seeds of some Myristicaceae plants contain about 57.39% solid oil and are mainly composed of Myristic acid and tetradecenoic acid, such that the relative average content of these two acids exceeds 90% [2, 3]. Since their seeds are rich in C14 fatty acid [4], which can be used as a condensing agent in medical, beauty, cosmetics, and other industrial products, they are considered a high-quality raw material [3–6], implying a significant economic opportunity in exploring the resource of Myristaceae species.

Because of the use of their oil and wood in industry, Myristicaceae plants have attracted increasing attention from researchers. The studies conducted on this family have mainly focused on their fatty acid content [2], chemical composition [7], and taxonomy [8–12]. We conducted a field survey for this family in Yunnan province, China and found that some species appear in small population with few individuals, such as only 1 *Horsfieldia pandurifolia* Hu (location: 99° 46.956′E, 23°12.521′N, altitude of 1 000m) individual has been found in the North of Xiaohei River valley, Shuangjiang county, Yunnan. The original large population of *H. pandurifolia* has been destroyed and only small isolated population exist in the upper of the distribution area [13]. so, it is important to conserve the species using the effective measure. However, the systematic position of some species within the family remains controversial, such as *H. pandurifolia.* Therefore, the precise identification and delimitation of the species was the key.

*H. pandurifolia* was first named by Hu in 1963 [14], and recorded as *H. pandurifolia* Hu in Flora Yunnanica [15], but the *H. pandurifolia* was incorporated into the *Horsfieldia Prainii* (King) Warb., and classified into *Horsfieldia macrocoma* (Miq) Warb. as a subspecies, and established genus *Endocomia* as a model species by de Wilde in 1984, namely *Endocomia macrocoma* ssp. *prainii* [9]. In 2004, Ye argued that the taxonomic boundaries of the genera *Horsfieldia* and *Endocomia* was not obvious, annulled *Endocomia*, restoring *H. macrocoma* [12], and in Flora of China (2008), the genus *Endocomia* was also annulled and *H. pandurifolia* was named as *H. prainii* [1]. In recent years, based on results of molecular biology approaches, the systematic position of *H. pandurifolia* was also differently suggested by Wu [8] and Cai [16]. it is argued about *H. pandurifolia* position, so it is necessary to study the relationship of Myristicaceae, especially for *H. pandurifolia.*

The chloroplast organelle is the most noticeable feature in plants, and its plastome is conserved than mitochondrial and nuclear genomes. Due to the low rate of nucleotide substitution, the chloroplast genome is used frequently in phylogeny studies [17], at the same time, the chloroplast genome can be provide large sequence information, so it serves as good candidates for high resolution DNA barcoding [16]. With the advent of sequencing technology such as Illumina, Nanopore, there are increasing reports of complete chloroplast genome for plant phylogenetic analysis, such as in Acanthoideae [18], Magnoliaceae [19], and Zingiberaceae [20] etc. including the phylogenetic relationships study on the Chinese *Horsfieldia* based on the chloroplast genome analysis [16]. However, the analysis of phylogenetic relationship for the species of Myristicaceae that distributed in China using chloroplast genome has not been reported.

In this study, we sequenced, assembled, and characterized the chloroplast genome of two species of Myristicaceae, *Knema globularia* (Lam.) Warb. and *Knema cinerea* (Poir.) Warb., using the Roche/454 sequencing platform. To analyze the organization, gene contents, patterns of nucleotide substitution, simple sequence repeats (SSRs), and phylogenetic relationships, the previously reported chloroplast genomes of eight Myristicaceae species were considered for comparative analyses with the two newly assembled chloroplast genomes. Our study aims were as follows: (1) evaluate the variations within the 10 species and the structural diversity of

the chloroplast genome among Myristicaceae species; (2) upgrade our understanding of the application value of the chloroplast genome of Myristicaceae and provide novel genetic resources for future research in Myristicaceae; and (3) to provide a molecular evidence for the taxonomic classification of Myristicaceae.

## Materials and methods

### Plant material and DNA extraction

The species used in this study were identified according to the records in Flora of China. The samples information, including the two newly sequenced species and eight reported species, were listed in the Table 1, and the species names in this table were derived from the Flora Reipublicae Popularis Sinicae and Flora of China. The collection sites included Nangunhe National Nature Reserve, Xishuangbanna National Nature Reserve and Yunnan Institute of Tropical Crops (YITC), Yunnan, China. All the live specimen was planting on site, so, the herbarium were not been made. Fresh leaves of Myristicaceae plants were sampled and immediately dried with silica gel, and total genomic DNA per germplasm was extracted from 100 mg dried leaves using the DNeasy Plant Mini Kit (QIAGEN, Valencia, CA, USA).

### DNA sequencing

DNA concentrations were quantified using Life Invitrogen Qubit® 3.0 (Life, Invitrogen, USA). The total genomic DNA per germplasm with over 30 ng µL$^{-1}$ were used for subsequent steps. Purified DNA (5 mg) was sheared by nebulization with compressed nitrogen gas, which yielded DNA fragments of 400–800 bp in length, with single strands of DNA being recovered by denaturing treatment after the modification of terminal repair and specific adaptor sequence connections. Specific proportions of single-stranded DNA libraries were immobilized on DNA capture magnetic beads for Emulsion PCR and then sequenced in GS FLX reagents. After reaction, 4–6 billion base information sites were obtained.

**Table 1. The information of ten samples used in this analysis.**

| Sample No. | Sample name | Collector number | Date of collection | GPS coordinates | Site of specimen preservation | Accession number in GenBank |
|---|---|---|---|---|---|---|
| 1 | *Horsfieldia amygdalina* (Wall.) Warb. | 20140443 | 2019-02-13 | 100˚47′E 22˚00′N | YITC | MK285561 |
| 2 | *Horsfieldia pandurifolia* Hu | 20090402 | 2019-02-13 | 100˚47′E 22˚00′N | YITC | MH445411 |
| 3 | *Horsfieldia kingii* (Hook.f.) Warb. | 20140404 | 2019-02-13 | 100˚47′E 22˚00′N | YITC | MK285562 |
| 4 | *Knema elegans* Warb. | 20140606 | 2019-02-26 | 100˚07.38′E 21˚52.82′N | Xishuangbanna National Nature Reserve | MK285564 |
| 5 | *Knema cinerea* (Poir.) Warb. | 20140469 | 2019-02-27 | 101˚13.18′E 21˚54.00′N | Xishuangbanna National Nature Reserve | MN683756 |
| 6 | *Knema furfuracea* (Hook.f. &Thomson) Warb. | 20140428 | 2019-02-25 | 100˚02.39′E 21˚13.77′N | Xishuangbanna National Nature Reserve | MK285563 |
| 7 | *Knema linifolia* (Roxb.) Warb. | 20140475 | 2019-02-13 | 99˚04.39′E 23˚16.43′N | Nangunhe National Nature Reserve | MN683753 |
| 8 | *Knema conferta* (King) Warb. | 20161101 | 2019-02-14 | 99˚01.013′E 23˚16.943′N | Nangunhe National Nature Reserve | MN683754 |
| 9 | *Knema globularia* (Lam.) Warb. | 20140487 | 2019-02-27 | 101˚33.77′E 21˚36.75′N | Xishuangbanna National Nature Reserve | MN683755 |
| 10 | *Myristica yunnanensis* Y.H.Li | 20140492 | 2019-02-13 | 100˚47′E 22˚00′N | YITC | MK285565 |

## Genome assembly and annotation

The chloroplast genome was assembled using CLC Genomic Workbench v3.6 (http://www.clcbio.com), the contigs were aligned (≥90% similarity and query coverage) and ordered, according to the reference chloroplast genome of *K.elegans* which was reported by author. The genes in the chloroplast genome were predicted using the Dual Organellar GenoMe Annotator (DOGMA) program [21]. The protein-encoding genes, location of the ribosomal RNA (rRNA) and transfer RNA (tRNA) genes were annotated using BLASTX and BLASTN searches. To accurately determine the boundaries between introns and exons and the starting and stopping codon positions of protein-coding genes, the annotated results were manually examined, and the codon position was adjusted by comparing them with homologous genes in the reference genomes, namely *H. pandurifolia*, *H. kingii* and *K. elegans*, based on phylogenetic proximity. The chloroplast genome map was drawn using Genome Vx software [22], and the chloroplast genome sequences of *K.globularia* and *K.cinerea* have been deposited to the GenBank, the accession numbers were listed in Table 1.

## Repeats analysis

Microsatellite Identification Tool (Misa-web-IPK Gatersleben (ipk-gatersleben.de)) [23] was used to identify the SSRs with the following parameters: 10 for mononucleotide, 5 for dinucleotides, 4 for trinucleotides, and 3 for more than 4-base SSR motifs. Long-repeat analyses of 10 species was done using the program REPuter (https://bibiserv.cebitec.uni-bielefeld.de/reputer) [24] involving the default parameters.

## Genome comparison

Performing comparative analysis for the 10 Myristaceae species, the chloroplast genome of *Liriodendron tulipifera* L. from the same order Magnoliales was downloaded from GenBank (NC_008326.1, https://ncbi.nlm.nih.gov/search/all/?term=NC_008326.1) and used as a reference. The sequence identity of 11 species chloroplast genomes were plotted using the mVISTA program with the LAGAN mode [25]. To detect the rearrangement and inverse evolutionary events, multiple genome alignments were conducted using the progressive mauve algorithm [26], as implemented in the Geneious software package (Biomatters, Auckland, New Zealand), in the Mauve options, change the alignment algorithm to MCM (the Mauve Contig Mover) alignment. The borders of small single-copy (SSC), large single copy (LSC), and inverted repeat (IR) regions among Myristicaceae species were visually displayed and compared using Irscope [27] based on these species annotation GenBank (.gb) files. To detect the hotspots of the intergenetic divergence, 74 coding sequences, and 41 intergenic space sequences were extracted for each species using Phylosuite ver. 1.1.152 [28] and aligned in batches using MAFFT [29], using the—auto = strategy and the codon alignment mode. Following this step, the nucleotide diversity (Pi) value was calculated for each of the 115 loci using DnaSP ver.6.12.03 [30]. To determine whether coding genes in 10 species had selective pressure during evolution, DnaSP software [30] was used to calculated synonymous (dS) and nonsynonymous (dN) values of coding genes of 10 species as well.

## Phylogenetic analysis

To obtain a more comprehensive results, 16 previously reported chloroplast genomes of Myristicaceae by Cai et al. [16] were also downloaded from GenBank (MN486685, MN486686, and MN495958–MN495971) and performed the phylogenetic analyses together with these 10 sequences in this study (the accession numbers are listed in Table 1), and using *L.tulipifera* as

outgroup. After the alignment of all sequences with MAFFT software, the phylogenetic analyses were performed using maximum likelihood (ML) and Bayesian inference (BI) with MEGA 7.0 [31] and MrBayes version 3.2.6 [32] respectively. Conditions were set as bellow: using the model of GTR+Γ for ML, and GTR+Γ+I for BI analysis, the node support values were given in the form of posterior probability (PP) and bootstrap value (BV), when perform the BI analysis, the Markov chain Monte Carlo (MCMC) was run for 2, 000, 000 generations with two parallel searches using four chains.

## Results

### Characteristics of Myristicaceae chloroplast genome

The 10 complete chloroplast DNA sequences ranged from 154,527 bp (*K. furfuracea*) to 155,923 bp (*M. yunnanensis*), and they all displayed a quadripartite structure typical of angiosperms, which consisted of an SSC (15,072–30,998 bp), an LSC (86,188–92,561 bp), and a pair of IRs (37,754–48,154 bp) (Fig 1 and Table 2). The base composition of the 10 chloroplast genome sequences were analyzed and counted, and the GC content ranged from 39.19% (*M.

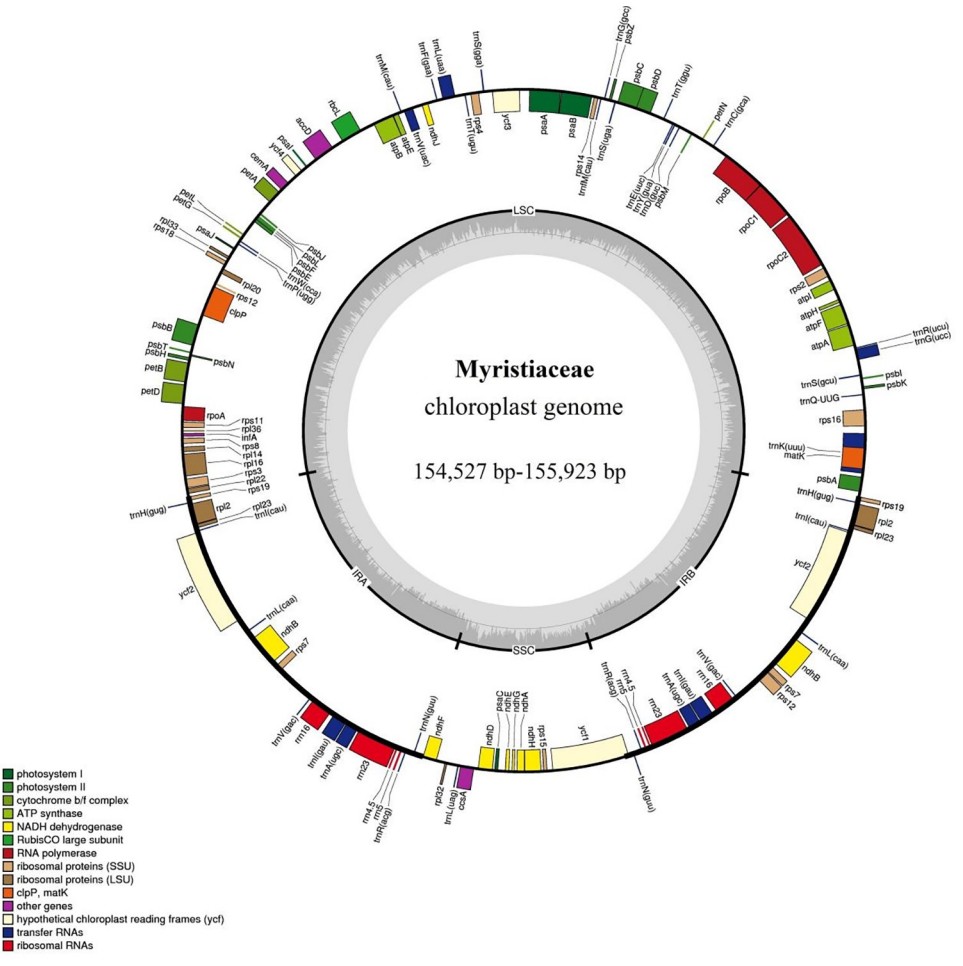

**Fig 1. Gene map of the Myristicaceae chloroplast genome.** Genes on the inside of the circle are transcribed in the counterclockwise direction, while genes on the outside of circle are transcribed clockwise. The different gene functional groups were labeled with different colors. In inner circle, the gray indicates GC content, while light gray represents AT content. LSC, Large Single Copy; SSC, Small Single Copy; IR, Inverted Repeat.

**Table 2. Summary of features of ten species of Myristicaceae.**

| genome features | H. amygdalina | H. pandurifolia | H. kingii | M. yunnanensis | K. furfuracea | K. elegans | K. linifolia | K. conferta | K. globularia | K. cinerea |
|---|---|---|---|---|---|---|---|---|---|---|
| cp length | 155,683 | 155, 695 | 155, 655 | 155, 923 | 154, 527 | 155, 691 | 155, 754 | 155, 744 | 155, 726 | 155, 690 |
| LSC length | 86,931 | 92, 561 | 86, 913 | 87, 088 | 86, 188 | 86, 883 | 86, 991 | 86, 926 | 86, 898 | 86, 881 |
| SSC length | 30,998 | 15, 072 | 20, 691 | 20, 731 | 20, 229 | 20, 686 | 20, 683 | 20, 770 | 20, 674 | 20, 761 |
| IR length | 37,754 | 48, 062 | 48, 052 | 48, 104 | 48, 110 | 48, 122 | 48, 080 | 48, 052 | 48, 154 | 48, 048 |
| Genes | 124 | 121 | 123 | 124 | 124 | 123 | 128 | 128 | 131 | 131 |
| CDS | 86 | 84 | 85 | 85 | 87 | 85 | 89 | 89 | 92 | 92 |
| tRNA | 27 | 26 | 27 | 27 | 27 | 27 | 31 | 31 | 31 | 31 |
| rRNA | 8 | 8 | 8 | 8 | 8 | 8 | 8 | 8 | 8 | 8 |
| GC% | 39.24 | 39.21 | 39.23 | 39.19 | 39.23 | 39.20 | 39.19 | 39.20 | 39.22 | 39.21 |

yunnanensis) to 39.24% (*H. amygdalina*), while the AT content ranged from 60.76% (*H. amygdalina*) to 60.81% (*M. yunnanensis*). The number of genes ranged from 121 to 131, with varied numbers of CDS, rRNA, and tRNA in different species (Table 3); among these genes, 11 genes (*trnQ-UUG, rps19, psbB, trnS-GGA, rpoB, atpH, rps7, trnV-GAC, ndhH, rpl23,* and *trnL-CAA*) had one intron, while *rps7* contained two introns in *H. amygdalina, K. elegans,* and *M. yunnanensis*.

## Repeat analysis

Tandem repeat sequences (TRSs) have an important influence in terms of gene structure, function and evolution, and so on [33]. SSRs are tandemly repeated motifs with a length of 1–6 bp, which have been widely used as molecular markers in evolutionary biology and population genetics [34, 35]. To explore the genetic changes evident in the Myristicaceae species analyzed, we performed tandem repeat and SSRs analysis. In this study, we identified an average of 62 SSR loci in the complete chloroplast genome of the studied species, including 45 mononucleotide SSR loci, 5 dinucleotide SSR loci, 2 trinucleotide SSR loci, 8 tetranucletide SSR loci, and 2 pentanucleotide SSR loci. In all species, only one hexanucleotide was found in *H. pandurifolia*, (AATAAA)3, located in the *matK~rps16* region (Fig 2A). SSR information, from trinucleotides to hexanucleotides, are displayed in Table 4, because of the large number of mononucleotide and dinucleotides SSR sites, they are listed in S1 Table.

At the same time, a total of 38 types of TRSs were detected in all species, with the repeat type ranging from 16 to 24 per species (Fig 2B); these repeats are mainly distributed in the *ycf2, trnV-GAC~rps7,* and trnN-*GUU~trnR-ACG* in the IR regions, the *rps11, ndhB, petN~psbM, atpH~atpI, rpoB~trnC-GCA, atpB~rbcL, trnP-UGG~psaJ, psbZ~trnG-GCC, rpl20~rps12, rpl32~trnL-UAG, trnC-GCA~petN, trnD-GUC~trnY-GUA,* and *trnT-UGU~trnF-GAA* in the LSC, and the *ycf1, ycf1~trnN-GUU, ndhD~ccsA, ccsA~ndhF,* and *rpl32~trnL-UAG* in the SSC. Among them, the shortest TRSs with a base sequence of TTTATATAA were detected in the species *K. cinerea, K. elegans, K. furfuracea,* and *K. linifolia*, while the longest TRS, with a base sequence of AGAAAAATGGAGACTATTTCTTTTTATTTAT was detected only in *K. linifolia* (Fig 3).

## Sequence divergence

To detect the selection pressures in protein coding genes (PCGs) in Myristicaceae chloroplast genomes, nonsynonymous (dN) substitutions, synonymous (dS) substitutions, and their ratios (dN/dS) were calculated, using 74 PCGs of 10 Myristicaceae species (Fig 4, S2 Table). Most PCGs had dN/dS values that were less than 1, and only 11 (*accD, ccsA, matK, ndhF, ndhG,*

**Table 3. Gene present in the chloroplast genome of Myristicaceae.**

| Category | Group of genes | Name of genes |
|---|---|---|
| Protein genes | Photosystem I | *psaA, psaB, psaC, psaI, psaJ, ycf3* |
| | Photosystem II | *psbA*[3(2)]*, psbB*[1,3(2)]*, psbC, psbD, psbE, psbF, psbH, psbI, psbJ, psbK, psbL, psbM, psbT, psbZ* |
| | Subunit of cytochrome | *petA, petB, petD, petG, petL, petN* |
| | Subunit of synthase | *atpA, atpB, atpE, atpF, atpH*[1]*, atpI* |
| | Large subunit of rubisco | *rbcL* |
| | NADH dehydrogenase | *ndhA, ndhB*[3(2)]*, ndhC, ndhD, ndhE, ndhF, ndhG, ndhH*[1]*, ndhI, ndhJ, ndhK* |
| | ATP dependent protease subunit P | *clpP* |
| | Chloroplast envelope membrane protein | *cemA* |
| RNA genes | Ribosomal RNA genes (rRNA) | *rrn5*[3(2)]*, rrn4.5*[3(2)]*, rrn16*[3(2)]*, rrn23*[3(2)] |
| | Transfer RNA genes (tRNA) | *trnH-GUG, trnQ-UUG*[1]*, trnS-GCU, trnR-UCU, trnC-GCA, trnD-GUC, trnY-GUA, trnE-UUC, trnT-GGU, trnS-UGA, trnG-GCC, trnS-GGA*[1]*, trnT-UGU, trnF-GAA, trnM-CAU*[3(3)]*, trnW-CCA, trnP-UGG, trnI-GAU*[3,4]*, trnL-CAA*[1,3]*, trnV-GAC*[1,3(2)]*, trnA-UGC*[3(2),4]*, trnR-ACG, trnN-GUU, trnL-UAG* |
| Ribosomal proteins | Small subunit of ribosome | *rps2, rps3, rps4, rps7*[1,2,3(2)]*, rps8, rps11, rps12*[3(2)]*, rps14, rps15, rps16, rps18, rps19* |
| Transcription | Large subunit of ribosome | *rpl2*[3(2)]*, rpl14, rpl16, rpl20, rpl22, rpl23*[1,3(2)]*, rpl32, rpl33, rpl36* |
| | DNA dependent RNA polymerase | *rpoA, rpoB*[1]*, rpoC1, rpoC2* |
| Other genes | Maturase | *matK* |
| | Subunit acetyl-coA carboxylase | *accD* |
| | C-type cytochrome systhesis | *ccsA* |
| | Hypothetical proteins | *ycf2*[3(2)]*, ycf4, ycf1* |
| | Translation initiation factor IF-1 | *infA* |

[1] Gene with one intron in all species except for *H. pandurifolia, H. kingii, K. furfuracea, K. linifolia, K. conferta, K. globularia* and *K. cinerea*

[2] Gene with two intron in *H. amygdalina, K. elegans* and *M. yunnanensis*

[3()] Gene with copies, the number in the brackets was the number of copies and

[4] Gene only distributed in *K. conferta, K. globularia, K. cinerea* and *K. linifolia*

*psaA, rbcL, rpoA, rpoC2, ycf1,* and *ycf2*) had dN/dS values greater than 1; among these 11 PCGs, the dN/dS ratio of the *psaA* gene in most species comparisons was greater than 1, while for the *rpoA* gene, the ratio was greater than 1 only in the comparison group of *Myristica* vs. *Knema* (Fig 4, S2 Table). Most of the PCGs had negative selection, and only a few PCGs had positive selection. The value of synonymous (dS) ranged from 0.00 to 0.68 (*psaA* gene) in all of the genes. Most PCGs, including *atpB, atpH, cemA, clpP, ndhK, petG, psaC, psbD, psbE, psbF, psbJ, psbK, psbL, psbM, psbT, psbZ, rpl2, rpl14, rpl23, rpl33, rpl36, rps11, rps15, rps16, rps19,* and *ycf3,* showed no nonsynonymous (dN) changes.

To clearly understand the nucleotide changes in the Myristicaceae chloroplast genome, we calculated the nucleotide diversity (Pi) values of the PCGs and intergenic spacers. In the coding region, the mean Pi in the PCGs was 0.00279 (ranging from 0 to 0.00886, Fig 5A). In total, 11 mutation sites (Pi > 0.005) were identified, including *matK, ndhD, ndhF, ndhG, psbL, rpl16,*

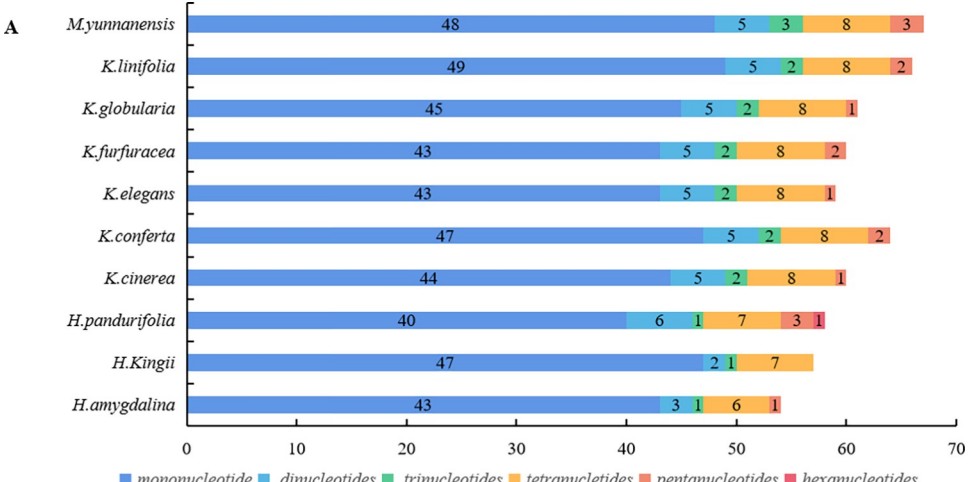

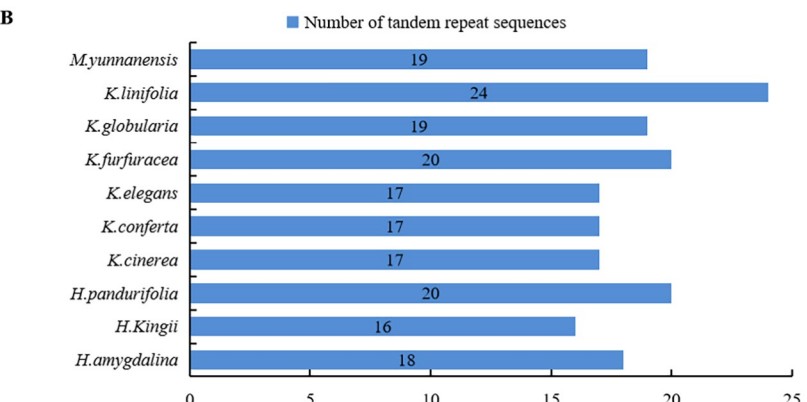

**Fig 2. Types and numbers of repeat in the chloroplast genomes of ten cp genome of Myristicaceae.** A, Numbers and types of SSRs; B, Numbers of Tandem repeat sequences.

*rpl32*, *rpoA*, *rps3*, *rps19*, and *ycf1*. In intergenic spacers, the mean Pi value was 0.006933 (ranging from 0.00054 to 0.04744, Fig 5B), and there were 18 mutation sites (Pi > 0.005) in these spacers, including *accD~psaI*, *atpF~atpH*, *matK~rps16*, *ndhC~trnM-CAU*, *ndhF~rpl32*, *petA~psbJ*, *petN~psbM*, *psbE~petL*, *psbM~trnD-GUC*, *rpl20~rps12*, *rpl32~trnL-UAG*, *rpoB~trnC-GCA*, *rps16~trnQ-UUG*, *trnC-GCA~petN*, *trnE-UUC~trnT-GGU*, *trnN-GUU~trnR-ACG*, *trnT-UGU~trnF-GAA*, and *ycf3~trnS-GGA*. We also assessed the statistics of initiation codon and termination codon of 74 PCGs, the results showed that the initiation codons were mainly ATG, GTG, CAA, ACG, CCA, AAT, GGG, and AAC, and the termination codons included TGA, TAG, and TAA; among these, ATG was used as the initiation codon in 63 genes, and these termination codons were used at a similar frequency.

## Genome comparison

To investigate the variation in chloroplast genome sequences, the 10 Myristicaceae species were compared with the mVISTA using the annotation sequence of *L. tulipifera*. The results indicated that the chloroplast genome sequences of Myristicaceae were relatively conserved, although a certain level of variation was detected. The pair of reverted repeat region was highly conserved than the LSC region and SSC region, and the PCGs were highly conserved than

**Table 4. SSR site information of ten species of Myristicaceae.**

| Species | Type | Repeat unit | Start | End | Location | Type | Repeat unit | Start | End | Location |
|---|---|---|---|---|---|---|---|---|---|---|
| *H. amygdalina* | tetra | (TTTG)3 | 33689 | 33700 | *atpH~atpI* | tetra | (ACAT)3 | 105492 | 105503 | *rpl22* |
| | tetra | (TCTA)3 | 55980 | 55991 | *psbC~trnS-UGA* | tetra | (CATT)3 | 145487 | 145498 | *ycf1* |
| | tetra | (TTTC)3 | 69141 | 69152 | *trnF-GAA~ndhJ* | penta | (AAATA)3 | 97116 | 97130 | *psbH~petB* |
| | tetra | (AATG)3 | 83317 | 83328 | *cemA* | | | | | |
| *H. Kingii* | tetra | (TTTG)3 | 59557 | 59568 | *atpH~atpI* | tetra | (AATG)3 | 109157 | 109168 | *cemA* |
| | tetra | (TCTA)3 | 81828 | 81839 | *psbC~trnS-UGA* | tetra | (CTTT)3 | 129381 | 129392 | *rpl16~rps3* |
| | tetra | (TAAC)3 | 93138 | 93149 | *trnT-UGU~trnF-GAA* | tetra | (ACAT)3 | 131314 | 131325 | *rpl22* |
| | tetra | (TTTC)3 | 95007 | 95018 | *trnF-GAA~ndhJ* | | | | | |
| *H. pandurifolia* | hexa | (AATAAA)3 | 28500 | 28517 | *matK~rps16* | tetra | (AATG)3 | 88391 | 88402 | *cemA* |
| | tetra | (TTTG)3 | 38712 | 38723 | *atpH~atpI* | tetra | (CTTT)3 | 108650 | 108661 | *rpl16~rps3* |
| | penta | (TTGAT)3 | 52817 | 52831 | *trnC-GCA~petN* | tetra | (ACAT)3 | 110583 | 110594 | *rpl22* |
| | penta | (AATCA)3 | 52842 | 52856 | *trnC-GCA~petN* | penta | (TATTT)3 | 142128 | 142142 | *ndhD~psaC* |
| | tetra | (TCTA)3 | 61093 | 61104 | *psbC~trnS-UGA* | tri | (AAT)4 | 148613 | 148624 | *rps15~ycf1* |
| | tetra | (TAAC)3 | 72349 | 72360 | *trnT-UGU~trnF-GAA* | tetra | (CATT)3 | 150631 | 150642 | *ycf1* |
| *K. cinerea* | tetra | (TTTG)3 | 38736 | 38747 | *atpH~atpI* | tetra | (AATG)3 | 88416 | 88427 | *cemA* |
| | tetra | (TCTA)3 | 61082 | 61093 | *psbC~trnS-UGA* | tetra | (CTTT)3 | 108655 | 108666 | *rpl16~rps3* |
| | penta | (ATTCT)3 | 72088 | 72102 | *trnT-UGU~trnF-GAA* | tetra | (ACAT)3 | 110591 | 110602 | *rpl22* |
| | tetra | (TAAC)3 | 72424 | 72435 | *trnT-UGU~trnF-GAA* | tri | (AAT)4 | 148593 | 148604 | *rps15~ycf1* |
| | tetra | (TTTC)3 | 74263 | 74274 | *trnF-GAA~ndhJ* | tetra | (CATT)3 | 150611 | 150622 | *ycf1* |
| | tri | (ATA)4 | 81105 | 81116 | *atpB~rbcL* | | | | | |
| *K. conferta* | penta | (ATTAA)3 | 37586 | 37600 | *atpF~atpH* | tetra | (AATG)3 | 88455 | 88466 | *cemA* |
| | tetra | (TTTG)3 | 38749 | 38760 | *atpH~atpI* | tetra | (CTTT)3 | 108689 | 108700 | *rpl16~rps3* |
| | tetra | (TCTA)3 | 61095 | 61106 | *psbC~trnS-UGA* | tetra | (CATT)3 | 150670 | 150681 | *ycf1* |
| | penta | (ATTCT)3 | 72091 | 72105 | *trnT-UGU~trnF-GAA* | tetra | (ACAT)3 | 110624 | 110635 | *rpl22* |
| | tetra | (TTTC)3 | 74287 | 74298 | *trnF-GAA~ndhJ* | tetra | (TAAC)3 | 72449 | 72460 | *trnT-UGU~trnF-GAA* |
| | tri | (ATA)4 | 81144 | 81155 | *atpB~rbcL* | tri | (AAT)4 | 148652 | 148663 | *rps15~ycf1* |
| *K. elegans* | tri | (AAT)4 | 13626 | 13637 | *rps15~ycf1* | tetra | (TAAC)3 | 93147 | 93158 | *trnT-UGU~trnF-GAA* |
| | tetra | (CATT)3 | 15644 | 15655 | *ycf1* | tetra | (TTTC)3 | 94986 | 94997 | *trnF-GAA~ndhJ* |
| | tetra | (TTTG)3 | 59459 | 59470 | *atpH~atpI* | tri | (ATA)4 | 101828 | 101839 | *atpB~rbcL* |
| | tetra | (TCTA)3 | 81805 | 81816 | *psbC~trnS-UGA* | tetra | (AATG)3 | 109139 | 109150 | *cemA* |
| | penta | (ATTCT)3 | 92811 | 92825 | *trnT-UGU~trnF-GAA* | tetra | (CTTT)3 | 129379 | 129390 | *rpl16~rps3* |
| *K. furfuracea* | tetra | (ACAT)3 | 131315 | 131326 | *rpl22* | tetra | (ACAT)3 | 37646 | 37657 | *rpl22* |
| | tetra | (TAAC)3 | 205 | 216 | *trnT-UGU~trnF-GAA* | tri | (AAT)4 | 75258 | 75269 | *rps15~ycf1* |
| | tetra | (TTTC)3 | 2044 | 2055 | *trnF-GAA~ndhJ* | tetra | (CATT)3 | 77276 | 77287 | *ycf1* |
| | tri | (ATA)4 | 8160 | 8171 | *atpB~rbcL* | penta | (ATTAA)3 | 119856 | 119870 | *atpF~atpH* |
| | tetra | (AATG)3 | 15471 | 15482 | *cemA* | tetra | (TTTG)3 | 121023 | 121034 | *atpH~atpI* |
| | tetra | (CTTT)3 | 35710 | 35721 | *rpl16~rps3* | tetra | (TCTA)3 | 143371 | 143382 | *psbC~trnS-UGA* |
| *K. globularia* | penta | (ATTCT)3 | 154374 | 154388 | *trnT-UGU~trnF-GAA* | tetra | (TAAC)3 | 93129 | 93140 | *trnT-UGU~trnF-GAA* |
| | tri | (AAT)4 | 13592 | 13603 | *rps15~ycf1* | tetra | (TTTC)3 | 94968 | 94979 | *trnF-GAA~ndhJ* |
| | tetra | (CATT)3 | 15610 | 15621 | *ycf1* | tri | (ATA)4 | 101817 | 101828 | *atpB~rbcL* |
| | penta | (ATTAA)3 | 58280 | 58294 | *atpF~atpH* | tetra | (AATG)3 | 109128 | 109139 | *cemA* |
| | tetra | (TTTG)3 | 59443 | 59454 | *atpH~atpI* | tetra | (CTTT)3 | 129366 | 129377 | *rpl16~rps3* |
| | tetra | (TCTA)3 | 81788 | 81799 | *psbC~trnS-UGA* | | | | | |
| *K. linifoia* | tetra | (ACAT)3 | 131301 | 131312 | *rpl22* | tetra | (TTTC)3 | 74378 | 74389 | *trnF-GAA~ndhJ* |
| | penta | (ATTAA)3 | 37598 | 37612 | *atpF~atpH* | tri | (ATA)4 | 81227 | 81238 | *atpB~rbcL* |
| | tetra | (TTTG)3 | 38761 | 38772 | *atpH~atpI* | tetra | (AATG)3 | 88538 | 88549 | *cemA* |
| | tetra | (TCTA)3 | 61144 | 61155 | *psbC~trnS-UGA* | tetra | (CTTT)3 | 108778 | 108789 | *rpl16~rps3* |
| | penta | (ATTCT)3 | 72151 | 72165 | *trnT-UGU~trnF-GAA* | tetra | (ACAT)3 | 110715 | 110726 | *rpl22* |
| | tetra | (TAAC)3 | 72533 | 72544 | *trnT-UGU~trnF-GAA* | tri | (AAT)4 | 148757 | 148768 | *rps15~ycf1* |

*(Continued)*

**Table 4.** (Continued)

| Species | Type | Repeat unit | Start | End | Location | Type | Repeat unit | Start | End | Location |
|---|---|---|---|---|---|---|---|---|---|---|
| *M. yunnanensis* | tetra | (CATT)3 | 150775 | 150786 | *ycf1* | penta | (ATTCT)3 | 93002 | 93016 | *trnT-UGU~trnF-GAA* |
| | tri | (AAT)4 | 13667 | 13678 | *rps15~ycf1* | tetra | (TAAC)3 | 93334 | 93345 | *trnT-UGU~trnF-GAA* |
| | tetra | (CATT)3 | 15685 | 15696 | *ycf1* | tetra | (TTTC)3 | 95203 | 95214 | *trnF-GAA~ndhJ* |
| | penta | (ATCCA)3 | 48367 | 48381 | *matK~rps16* | tri | (ATA)4 | 102041 | 102052 | *atpB~rbcL* |
| | penta | (GAATA)3 | 52032 | 52046 | *trnS-GCU~trnR-UCU* | tetra | (AATG)3 | 109358 | 109369 | *cemA* |
| | tetra | (TTTG)3 | 59658 | 59669 | *atpH~atpI* | tetra | (CTTT)3 | 129630 | 129641 | *rpl16~rps3* |
| | tetra | (TCTA)3 | 82020 | 82031 | *psbC~trnS-UGA* | tetra | (ACAT)3 | 131556 | 131567 | *rpl22* |
| | tri | (TAA)4 | 91499 | 91510 | *ycf3~trnS-GGA* | | | | | |

non-coding sequences, particularly in intergenic regions. The intergenic spacer region featured a high level of variation within the genome, including *atpH~atpI*, *trnS-GGA~rps4*, *ndhC~trnM-CAU*, *petA~psbJ*, *trnP-UCG~psaJ*, *psbE~petL*, and *psbH~petB*, and the protein coding genes were *ycf2*, *rpoC1*, *clpP*, and *rps12* (Fig 6).

Previous studies have found that the plastome sequence was conserved in flowering plants [36], although, due to evolutionary events, changes occurred in the size and boundaries of individual replicates and reverse repeats [37, 38]. Comparison among the reverse repeat region, the LSC region, and the SSC region boundaries in the 10 species of Myristicaceae are presented in Fig 7. Most species exhibit some variation in the number of nucleotides in the boundaries of the LSC, IR, and SSC regions. Except *H. pandurifolia*, the studied Myristicaceae species had the same set of genes at the border: the *rpl22* and *trnH* genes were located in the LSC region, the *ndhF* and *trnR* genes were located in the SSC region, and the *rps19* and *rrn5* genes were found in the IR regions. However, for *H. pandurifolia*, the *trnH* gene was located in the IR regions, the *rps19* gene was located on the LSC/IRs border, and the positioning of the *ycf1* gene in the IRb/SSC border was observed only in the genome of *H. pandurifolia*.

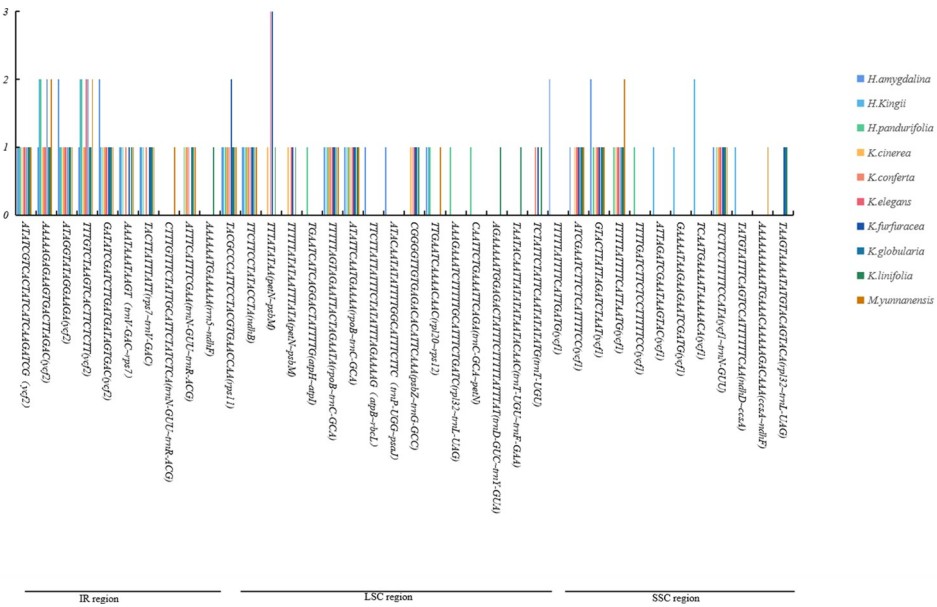

**Fig 3. Details of tandem repeats sequence in 10 species and number of repeats in each species.** The gene or intergenic regions in parentheses indicates the location of the tandem repeat. The ordinate represents the number of repetitions.

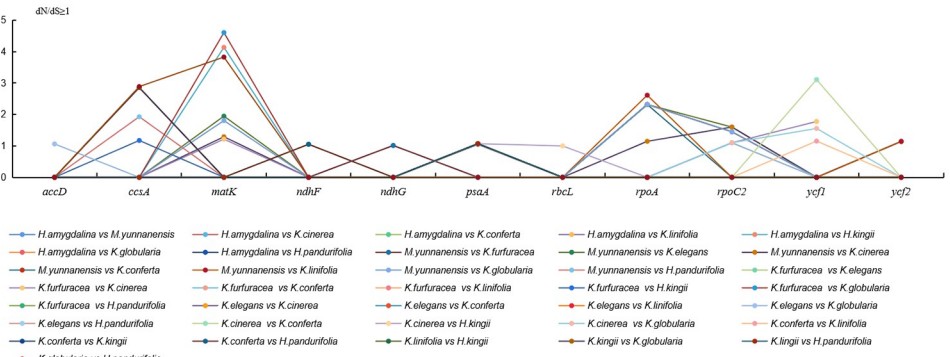

**Fig 4. The genes of dN/dS>1 from 10 cp genomes of Myristicaceae.**

## Phylogenetic analysis

To determine the phylogenetic relationship of the 26 samples of Myristicaceae, phylogenetic trees were reconstructed using the chloroplast genome of those species using BI and ML, taking *L. tulipifera* as the outgroup. The results showed that BI and ML analyses were congruent with high-support PP, 1.0, and MP, 100, in most relationships (Fig 8). All species of Myristicaceae were clustered in one clade with high support. Both analysis methods showed that *Horsfieldia* genus formed a single clade except for *H. pandurifolia* with high bootstrap values of 100%/1.0. In contrast, all *H. pandurifolia* samples were separated from *Horsfieldia* genus and formed a clade with bootstrap values of 100%/1.0 and formed a sister group with genus *Knema* and *Myristica*.

## Discussion

Myristicaceae plants are recognized as source plants of medical, wood, and oil products. *M. yunnanensis* is listed as a protected plant in China and is now considered endangered [39].

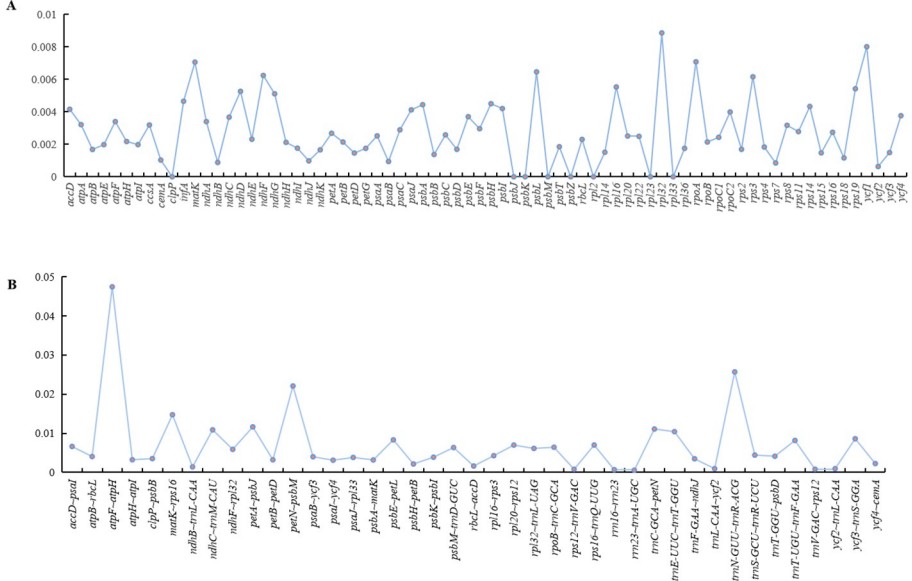

**Fig 5. Nucleotide diversity (Pi) in the chloroplast genome of ten Myristicaceae species.** (A:Pi value (range from 0–0.00886) of protein coding genes; B:Pi value (0.00054–0.04744) of intergenic regions genes).

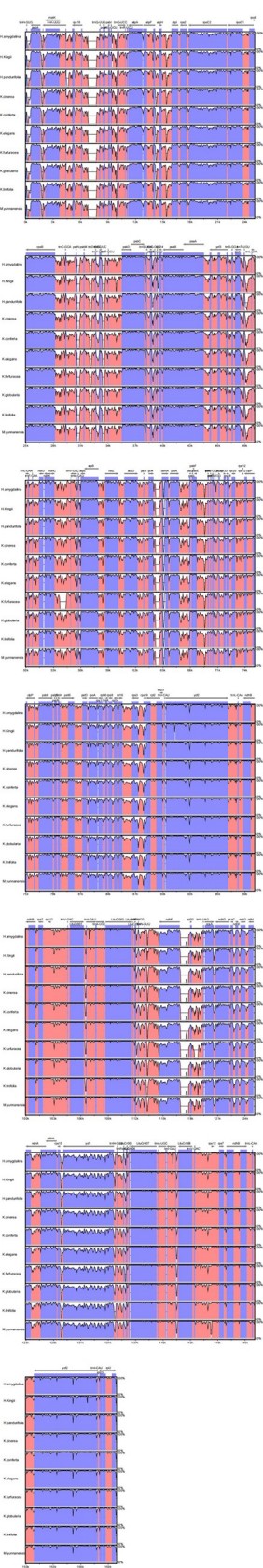

**Fig 6. Sequences alignment of ten chloroplast genome in the Myristicaceae family performed with mVISTA using annotation of *L. tulipifera* as reference.**

Until recently, the reports of genome sequences in this family had only been published by the authors [40–47] and Cai [16]. In this study, we comprehensively analyzed the chloroplast genome of Myristicaceae species and reconstructed phylogenetic tree using 26 genome sequences. The results showed that all genomes exhibited a quadripartite structure, including LSC, small SSC, and a pair of IR regions (IRa and IRb), which consisted of 84–92 PCGs, 26–31 tRNAs, and 8 rRNAs, with genome sizes that ranged from 154,527 to 155,923 bp and with GC % contents that ranged from 39.18% to 39.24%. Compared to species from the same order Magnoliales, the organization and structure of these 10 chloroplast genomes were similar to that of the *L. tulipifera* [48].

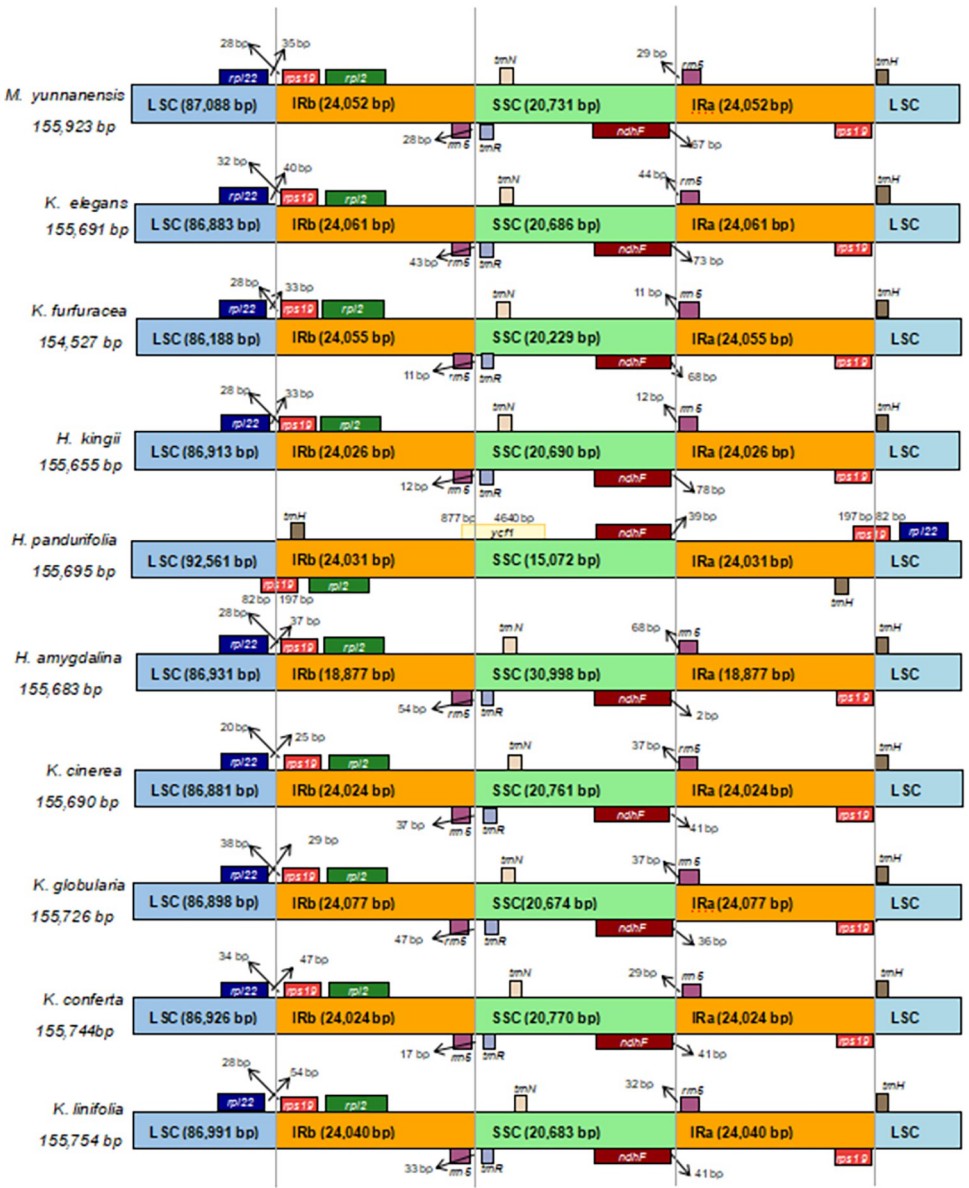

**Fig 7. Comparison of the borders of the IR, SSC and LSC regions among ten chloroplast genome of Myristicaceae.**

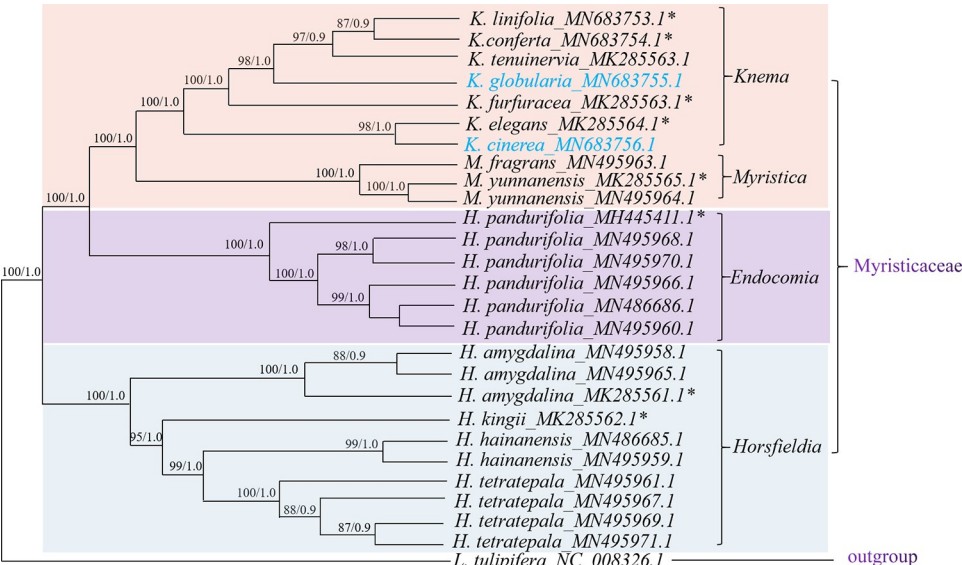

**Fig 8. The phylogenetic tree based on complete chloroplast genome sequences of Myristicaceae with ML and BI method.** Support values are bootstrap values (>50%, before slash) and posterior probability (>0.5, after slash), respectively. The species with blue font indicates the two newly sequenced in this study; "*" indicates that the data were derived from the authors.

Genome repeats play an indispensable role in gene expression, transcriptional regulation, chromosome construction, genomic structural variation, expansion, and rearrangement [49–51]. Similarly, we analyzed the repeat sequences of Myristicaceae, and the results are consistent with previous studies that most SSRs were located in intergenic spacers regions, followed by coding regions, and most TRSs were located in *ycf* genes and non-coding regions [19, 52–54]. The cpSSR has been widely used in phylogenetic evaluation and population genetics [55], and it is a very effective marker as well [56, 57]. In the chloroplast genome of Myristicaceae, mononucleotide repeats made up over 79%, and over 95% of the mononucleotide consisted of A or T bases, and the majority of SSRs were found in the SSC and LSC regions, the proportion of mononucleotide repeats and the base composition were similar to previous research [53, 55].

The variation in the chloroplast genome size was a result of the contraction and expansion of the reverse repeats (IRs) [58]. This contraction and expansion were observed in the chloroplast genome sequences of Myristicaceae. The size of the IRs ranged from 18,877 bp (*H. amygdalina*) to 24,077 bp (*K. globularia*). Despite the similar lengths of the IR regions of *H. pandurifolia* in relation to other Myristicaceae species, with the exception of *H. amygdalina*, some level of expansion and contraction were observed. Due to the different positions of the genes *rps19*, *ycf1*, and *trnH*, three types of variation in the border of IR-SC region appeared among these species, as a result of the contraction and expansion of reverse repeats. Type I occurred in *H. pandurifolia*, where parts of the genes *rps19* and *ycf1* were located in the IR region, and other parts were located in the LSC and SSC regions, respectively. Type II occurred in all other species, except *H. pandurifolia*, that the two *rps19* were located in the IR regions. Type III was found in all the studied species, except *H. pandurifolia*, that *trnH* was located in the LSC region whereas *trnH* was located in IR regions of *H. pandurifolia*. All genomes had *ndhF* in the SSC region, *rpl2* in the IR region, and *rpl22* in the LSC region.

dN, dS, and dN/dS were calculated to evaluate sequence diversity and purifying selection in species evolution. The results indicated low sequence diversity in most genes. dN/dS analysis showed that most protein coding genes faced negative selection (dN/dS < 1), only 11 genes

(*accD*, *ccsA*, *matK*, *ndhF*, *ndhG*, *psaA*, *rbcL*, *rpoA*, *rpoC2*, *ycf1*, and *ycf2*) were positively selected, and the genes with positively selected in this study have been reported in *Barleria prionitis* [18] and *Rheum* species [59]. Nucleotide diversity can be used to measure mutations in the population and can also be used to estimate evolutionary relationships [60]. In recent years, DNA barcoding has been considered as a reliable tool for resolving phylogenetic relationships and species authentication [53, 61]. In this study, we aligned the chloroplast genomes and found 11 and 18 mutation sites in the protein coding region and the intergenic spacer, respectively. These mutation hotspots provided valuable markers for identification of species as well as resolving phylogenetic relationships in the family.

Comparative genome analysis using mVISTA showed that the genomes are relatively conserved with minor variation, which mainly occurred in non-coding regions. Through the results of alignment, there were no considerable rearrangement being detected in the chloroplast genome.

The genome sequences were the effective resource for inferring phylogenetic relationships among species [62]. The phylogenetic relationship among the species of Myristicaceae is controversial, particularly *H. pandurifolia* [8, 12, 16]. In this study, the phylogenetic relationships of 26 Myristicaceae samples were inferred using ML and Bayesian methods. The genus *Knema* was clustered with the genus *Myristica*, and same-genus species were clustered in the same clade with a high support value, this was consistent with previously reported results which *Knema* species was found to share sister relationship with *Myristica* species [63]. In *Horsfieldia*, *H. kingii*, *H. hainanensis*, *H. tetratepala* and *H. amygdalina* were clustered together, however, *H. pandurifolia* was separated from the four *Horsfieldia* species forming a single clade, same as the result of Amplified Fragment Length Polymorphism (AFLP) [8]. For the morphological observation, the aril's color of *H. pandurifolia* was recorded as orange in Flora of China, whereas bright red as observed by Wu [8]; the seed apex of other *Horsfieldia* species were round, whereas *H. pandurifolia* was pointed, same as E. macrocoma spp. prainii; the testa color of *H. pandurifolia*, same as *E. macrocoma*, was variegated, whereas other *Horsfieldia* species' testa were brown [8, 13]. Based on the results of fatty acid research, Wu believed that *H. pandurifolia* should be separated from genus *Horsfieldia* [8]. In summary, molecular, morphological, and fatty acid data show that *H. pandurifolia* should be treated as a genus [8, 9].

In current study, the chloroplast genome sequences that used to analysis the phylogenetic relationship included all species except for *Myristica cagayanensis* Merr. And *Myristica simiarum* A.DC. distributed in China, which can better explain the phylogenetic relationships of the Chinese species of this family. This study suggests taxonomical revision that *H. pandurifolia* should be separated from the genus *Horsfieldia* and placed in the genus *Endocomia* (1984) proposed by W.J.de Wilde [9]. In addition, morphological observations of genus *Knema* by the authors showed that the flowers and leaves of *K. cinerea* and *K. elegans* are similar. Based on the phylogenetic tree, *K. furfuracea* and *K. linifolia* had a relatively close phylogenetic relationship, as the same result occurred in both *K. cinerea* and *K. elegans*. We will do further research in the future on the phylogenetic relationship of genus *Knema*.

## Conclusion

In this study, we sequenced two chloroplast genomes of *Knema* species, *K. globularia* and *K. cinerea*, and compared them with those of eight reported species. The size, gene content and the structure of Myristicaceae chloroplast genomes were similar, and comparative analysis revealed no gene inversion or relocation among chloroplast genome. We identified eleven highly variable sites, with seven intergenic spacer region and four PCGs, that could be explored to create valuable genetic markers for species authentication and phylogeny in Myristicaceae.

Through the comparison of the genomes, we identified 11 genes and 18 intergenic spacer regions were positively selected which can be used to analyze the population genetic structure of Myristicaceae. We comprehensively analyzed the phylogenetic relationship of Myristicaceae using 26 samples from 3 genera distributed in China. We suggesting taxonomical revision that *H. pandurifolia* should be separated from *Horsfieldia* and placed in the genus *Endocomia*.

## Supporting information

**S1 Table. Mononucleotide and dinucleotides SSR site information of ten species of Myristicaceae.**
(XLSX)

**S2 Table. The synonymous (dS) and dN/dS ratio values of 74 protein coding genes from ten cp genomes of Myristicaceae.**
(XLSX)

## Acknowledgments

We are grateful to thank Nangunhe National Nature Reserve and Xishuangbanna National Nature Reserve for the help of collection material.

## Author Contributions

**Data curation:** Changli Mao.

**Formal analysis:** Yu Wu.

**Funding acquisition:** Yu Wu.

**Investigation:** Changli Mao, Fengliang Zhang, Tian Yang.

**Methodology:** Changli Mao, Xiaoqin Li.

**Project administration:** Yu Wu.

**Resources:** Fengliang Zhang, Xiaoqin Li, Tian Yang, Qi Zhao, Yu Wu.

**Writing – original draft:** Changli Mao.

**Writing – review & editing:** Changli Mao, Yu Wu.

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
