## [Decision Letter · Decision Letter 0]

25 Jul 2022

PONE-D-22-11654Complete chloroplast genome sequence s of 10 Myristicaceae species with the comparative chloroplast genomics and phylogenetic relationships among themPLOS ONE

Dear Dr. mao,

Thank you for submitting your manuscript to PLOS ONE. After careful consideration, we feel that it has merit but does not fully meet PLOS ONE’s publication criteria as it currently stands. Therefore, we invite you to submit a revised version of the manuscript that addresses the points raised during the review process. The reviewers point out several. largely overlapping, areas of concern. These include the inclusion of data from previously published studies, greater clarity regarding methods, more in depth discussion of the results compared to precious hypotheses and character systems, and improved specimen metadata. Please address these concerns; these changes could greatly improve the manuscript's reach.

We look forward to receiving your revised manuscript.

Kind regards,

Michael Scott Brewer, Ph.D.

Academic Editor

PLOS ONE

Journal Requirements:

 “This work was supported by the National Natural Science Foundation of China (No.31960289)”

Reviewers' comments:

Reviewer's Responses to Questions

**Comments to the Author**

1. Is the manuscript technically sound, and do the data support the conclusions?

Reviewer #1: Partly

Reviewer #2: Yes

2. Has the statistical analysis been performed appropriately and rigorously? 

Reviewer #1: No

Reviewer #2: Yes

3. Have the authors made all data underlying the findings in their manuscript fully available?

Reviewer #1: Yes

Reviewer #2: Yes

4. Is the manuscript presented in an intelligible fashion and written in standard English?

Reviewer #1: No

Reviewer #2: Yes

5. Review Comments to the Author

Reviewer #1: The authors made a progress to our understanding to the plastome evolution and phylogeny of Myristicaceae. Because of poor support values for the systematic position of H. pandurifolia, you can not make a confident conclusion that Endocomia should be re-established. Detailed discussions on Endocomia should be given, mentioning and comparing studies based on phylogenetic analyses and morphological observations.

A great defect to your study is the neglection of many plastomes of Myristicaceae recently published by Cai et al. (2021) [doi: 10.1111/jse.12556]. More samples would be of great importance to resolve the systematic position of H. pandurifolia and to find new relationships. Additionally, phylogenetic analyses should also combine the data of Myristicaceae provided by PPA II (Li et al. 2021; doi: 10.1186/s12915-021-01166-2].

I also suggest the authors compare your phylogenetic results to Myristicaceae in an updated Chinese vascular plant tree of life (Hu et al. 2020; doi: 10.1111/jse.12642).

In general, I suggest the authors use as many data as possibly, not only yours but also those provided in previous studies.

Reviewer #2: The authors studied the chloroplast genomes of 10 Myristicaceae species, including 2 newly assembled for this study and 8 previously reported, aiming to provide potential genomic resources of this family and molecular evidence for taxonomical revision. The authors contribute to the taxonomy of Myristicaceae using modern technology, bringing novel insights to the science community. The study is very interesting to me. However, before the manuscript is ready for publishing, the following issues must be fixed.

Major issues:

1) The study aimed to provide evidences for the acceptance of the genus Endocomia proposed by W.J. de Wilde who validly published this genus name in 1984. Yet, the manuscript did demonstrate neither in-depth study on the nomenclature and classification of the family Myristicaceae, nor clear presentation on the inconsistency on the morphological description between floras and their own observations. It is not enough by just stating "we found that the character description of a few species was inconsistent with flora records in China as well" (L233-235) in the introduction and "morphological observations of Knema genus by the authors showed that the flowers and leaf of K. cinerea and K. elegans are similar" (L480-481) in the discussion. My recommendation is that the authors should spend more efforts in presenting the inconsistency and similar morphology clearly, which may increase the merit and interests of the study. The authors may also refer to online databases like Tropicos [https://www.tropicos.org/] and Plants of the World Online [https://powo.science.kew.org/] for the validity of scientific names and the current classification preferences, respectively.

2) The study tried to elucidate phylogenetic relationships of Myristacaceae. Yet, vouchering procedures were not clearly shown in the manuscript, particularly Table 1. If you collected specimens for the studied materials, it is a must to present in the manuscript as vouchers. Table 1 should also include collector number, date of collection and deposition (herbarium or institution) for your vouchers. GPS coordinates are recommended to be presented in a single column. A map visualising the distances of collection sites is a bonus. Digitised specimens are recommended to be submitted as supporting information as your proof of authentication.

3) Methodology is incomplete. Methods is missing for dN/dS analysis (i.e. which software you used?) and the authentication of studied materials (which flora(s) you referred to? any personal judgement(s)?).

4) The claims of this study is valid, supported by the clear results of phylogenetic analysis. Horsfieldia pandurifolia exhibited distinct phylogenetic position from the other two sistered Horsfieldia species with great support values. Yet, the introduction scarcely mentioned the application of chloroplast genomes in resolving phylogeny of Myristacaceae AND other angiosperms. Apart from the studies achieved by the authors in the past, the studies of Cai et al. (2021) should be mentioned and discussed. Meanwhile, any phylogenetic studies of Myristacaceae, like the one of Swetha et al. (2019), have been previosly reported? A brief summary is needed.

Cai, C. N., Ma, H., Ci, X. Q., Conran, J. G., & Li, J. (2021). Comparative phylogenetic analyses of Chinese Horsfieldia (Myristicaceae) using complete chloroplast genome sequences. Journal of Systematics and Evolution, 59(3), 504-514.

Swetha, V. P., Sheeja, T. E., & Sasikumar, B. (2019). DNA barcoding to resolve phylogenetic relationship in Myristica spp. Journal of Spices & Aromatic Crops, 28(2).

5) Personally speaking, I think it is a serious problem that the authors did not mentioned the species name of the two newly assembly chloroplast genomes right in the beginnings (abstract and introduction). Meanwhile, I can find no indication on the phylogenetic trees. Such practices negatively affected the clarity of the manuscript, as I do not know which species was newly assembled in this study, which one was previously reported. Simply stating the accession numbers on L276, 307-308 is not clear enough.

Minor issues:

6) Rooms of improvement for the presentation of figures are identified as the followings.

Fig. 3: Missing information as a few characters are invisible as "(trnD-..". The unit of y-axis should be adjusted (should be integer).

Fig. 4: There are many groups of comparisons between species, that the lines and data markers are coloured in similar tones. Frankly speaking, I cannot even read this figure accurately. My suggestion to use distinct colours, and change the shape of data markers instead of solely using dots. For example, use dots as the data markers for H. amygladina vs other species, use triangle for M. yunnanensis vs other species, etc. Fig. 4 should be presented as a single figure with indication (A,B,C) or otherwise three seperated figures. Also, please replace "ration" by "ratio" in the title.

Fig. 5: You should put a line to indicate the threshold of nucleotide diversity as you mentioned in the manuscript. The y-axis must be annotated (Pi). The unit of y-axis should be adjusted (Fig. 5A - 0.001) and (Fig. 5B - 0.005).

Fig. 7: Type I, II & III variation mentioned in the results should be visualised.

Fig. 8: Markers should be put to indicate the two newly assembled chloroplast genomes for this study. Meanwhile, accession number of each genome should be shown together with the species name. The sub-clades mentioned in the results should be visualised.

7) Please be aware that PLOS uses the numbered citation (citation-sequence) method for in-text citation, and the reference style is Vancouver for the reference list.

8) Gene names should be written in italic. Be aware of the presentations in Table 3 and 4.

9) Extensive language proofreading is highly recommended. Be aware of plural and singular forms of nouns, which affects the subsequent form of verbs. Some vocabularies were incorrectly used, conveying different meanings.

Other comments and suggestions for the Authors:

L198-199: … Comparing the genome structure of …

L202 and elsewhere: “sequence divergence analysis” but not “sequences divergence analysis”

L203 and elsewhere: “intergenic spacers” but not “gene spaces” or “gene spacers”

L205-206: … formed a sister clade with Myristica species supported by both high maximum likelihood bootstrap values and Bayesian posterior probabilities;

L219-220: … Since their seeds are rich in C14 fatty acid (Mao et al., 2017) [a space was missed between "acid" and in-text citation]

L233: significant economic opportunity in exploring the resource of Myristaceae species.

L232-233: We conducted a field survey for this family in [region/ provinces] and found that some species appear in small population with few individuals [could you provide an approximate number of individuals?].

L237-239: … platform. To analyze the organization, gene contents, patterns of nucleotide substitution, simple sequence repeats (SSRs), and phylogenetic relationships, the previously reported chloroplast genomes of eight Myristicaceae species were considered for comparative analyses with the two newly assembled chloroplast genomes.

L243: “novel” instead of “abundant”?

L254: suggest to removing the “Note” statement and put in main text

L257: USA). The total genomic DNA per germplasm with over 30 ng μL-1 were used for subsequent steps.

L272-274: … homologous genes in the reference genomes [state the species and accession number here], based on phylogenetic proximity.

L275: please clearly present the species name for “these species” here

L285-287: Performing comparative analysis for the 10 Myristaceae species species, the chloroplast genome of Liriodendron tulipifera (accession number) from the same order Magnoliales were downloaded from the public database NCBI GenBank and used as a reference.

L289: rearrangement and inverse evolutionary events …

L294 and elsewhere: “coding sequences” instead of “code sequences”

L300-308: This paragraph should be rewritten. MAFFT is used for alignment before constructing ML and BI trees. “Taxonomic positions” instead of “taxonomic locations”. The scientific name should be provided for the eight previously reported chloroplast genomes before the accession number, and they should be mentioned in the beginning of the manuscript.

L312-313 and elsewhere: quadripartite structure typical of angiosperms

L313-314: please check the size of the largest LSC, I found 92,561 bp in table 1

L314: 48,154 bp

L318-319: The number of genes ranged from 121 to 131, with varied numbers …

L319: suggest deleting “unique”

L320: suggest deleting “-fragment”

L325: K. conferta and (a space between "conferta" and “and” was missed)

L325-326: K. globularia should be italic

L326: Gene with copies? The number of copy (i.e. 1,2,3) should be clearly indicated in this table.

L335 and elsewhere: “the studied species” instead of “all species”

L338: please double check Table 4. I cannot find hexanucleotide (AATAAA)3 for H. pandurifolia in table 4, but H. kingii. If necessary, please revise the presentation of the table e.g. adding cell borders.

L342: genew??

L342,345,346: suggest replacing “genes or intergenic regions” by “in”.

L342 and elsewhere: “IR regions” instead of “IR region”

L346-347: “Among them” instead of “In these sequences”

L348: …, while the longest TRS …

L352: suggest removing the “Note” statement. If feasible, you may choose to show the information of mono- and di-nucleotides SSRs sites in supporting information.

L357,359,361,363 and elsewhere talking about “Sequence Divergence”: “PCGs” instead of “genes”

L363: “Most PCGs” instead of “Some genes”

L367: we calculated the nucleotide diversity (Pi) values of the PCGs and intergenic spacers.

L372-375 and elsewhere: you used “~” to present intergenic spacers in previous paragraphes. Please keep consistency. Choose either “~” or “-”.

L379: …, and the 3 termination codons …

L384: … Myristicaceae were relatively conserved.

L385 and elsewhere: “inverted repeat regions” but not “reverse repeat region”

L386: you already abbreviated “protein coding genes” as “PCGs”. Please use the abbreviation here.

L387: "highly" instead of “more highly”

L397: SSC regions. Except H. pandurifolia, the studied Myristicaceae species had the same set of genes at the border: the rpl22 …

L400: please add a description on the differences of H. pandurifolia here.

L400: what kind of “variation”? Any range?

L402: “LSC/IRs” but not “LSC/IRb”, since rps19 was also find in the border “LSC/IRa”

L405 and elsewhere: “phylogenetic trees” but not “phylogenetic tree”, since you have both ML and BI trees.

L409: sub-clade 1 should be marked in Fig. 8

L411: In the sub-clade 2, …

L413: “source plants” but not “producers”

L414: In-text citation is needed for this sentence.

L420: “consisted of” but not “included”

L422: … Compared to the species from the same order Magnoliales,

L423: “were” not “was”

L428: …, and the results are consistent with previous studies that most SSRs were

L429: … were located in ycf genes

L430: delete “, consistent with other research”

L434 and elsewhere: “previous studies” instead of “other research”

L436: “majority of” but not “major”

L443-446: please double check this sentence. Is the reason correct?

L446: Type I, II and III should be visualised in Fig. 7

L447-448: … LSC and SSC regions, respectively. In contrast, only …

L448: I cannot see “only the gene ycf1 was near the border” in Fig. 7

L449-452: except H. pandurifolia, that the two rps19 were located in the IR regions. Type III was found in all the studied species, except H. pandurifolia, that trnH was located in the LSC region whereas trnH was located in IR region of H. pandurifolia.

L458: what is the results reported in other species? Please describe.

L460: In-text citation is needed for the sentence “… estimate evolutionary relationships”.

L460-462: …, DNA barcoding have been considered as a reliable tool for resolving phylogenetic relationships and species authentication.

L464: “mutation hotspots” but not “genes and gene spacers”

L465: phylogenetic relationships

L466-467: Comparative genome analysis using mVISTA showed that the genomes are relatively conserved with minor variation, …

L468-469: …, there were no considerable rearrangement being detected in the chloroplast genomes.

L470: “phylogenetic relationships” but not “phylogenetic analysis”

L472: … 2015). In this study, the…

L473: “phylogenetic relationships” but not “phylogenetic tree”

L474 and elsewhere: “The genus XXX” but not “The XXX genus”

L476-477: … separated from the two Horsfieldia species forming a single clade.

L479-480: this study suggests taxonomical revision that H. pandurifolia should be separated from the genus Horsfieldia and placed in the genus Endocomia (1984) proposed by W.J. de Wilde [in-text citation of the protolgue is needed here].

L481: “leaves” but not “leaf”

L486: In this study, we sequenced two chloroplast genomes of Knema species, [species names needed], and compared them with those of eight reported species.

L487: "genomes" but not "genome"

L489: how many of “some”? Please provide a number instead of just saying “some”.

L489: “explored” instead of “developed”

L493-495: … a single clade, suggesting taxonomical revision that H. pandurifolia should be separated from Horsfieldia and placed in the genus Endocomia.

6. PLOS authors have the option to publish the peer review history of their article (what does this mean?). If published, this will include your full peer review and any attached files.

Reviewer #1: No

Reviewer #2: **Yes: **Kwan Ho WONG

---

## [Author Response · Author response to Decision Letter 0]

1 Sep 2022

Dear Editors and Reviewers:

Thank you for your letter and for the reviewers’ comments concerning our manuscript entitled “Complete chloroplast genome sequence s of 10 Myristicaceae species with the comparative chloroplast genomics and phylogenetic relationships among them” (ID: PONE-D-22-11654). Those comments are all valuable and very helpful for revising and improving our paper, as well as the important guiding significance to our researches. We have studied comments carefully and have made correction which we hope meet with approval. Revised portion are marked in red in the paper. The revisions were addressed point by point in the separate file labeled 'Response to Reviewers'.

---

## [Decision Letter · Decision Letter 1]

26 Sep 2022

PONE-D-22-11654R1Complete chloroplast genome sequence s of 10 Myristicaceae species with the comparative chloroplast genomics and phylogenetic relationships among themPLOS ONE

Dear Dr. mao,

Thank you for submitting your manuscript to PLOS ONE. After careful consideration, we feel that it has merit but does not fully meet PLOS ONE’s publication criteria as it currently stands. Therefore, we invite you to submit a revised version of the manuscript that addresses the points raised during the review process. The reviewers have additional suggestions/concerns regarding the revised submission. Reviewer 1 does not think that their criticisms were adequately addressed. They still has concerns regarding the addition of samples to the plastid phylogeny to provide a more holistic view of the relationships within the study system. Reviewer 2 outlined more minor issues that should be addressed.

Additionally, I ask you to include greater detail regarding the methods. The version numbers of all software used should be indicated; full details regarding the parameters used in each analysis should be included; and the data should be deposited in publicly accessible repositories. Much more information is require to reproduce the analyses presented here.

We look forward to receiving your revised manuscript.

Kind regards,

Michael Scott Brewer, Ph.D.

Academic Editor

PLOS ONE

Reviewers' comments:

Reviewer's Responses to Questions

**Comments to the Author**

1. If the authors have adequately addressed your comments raised in a previous round of review and you feel that this manuscript is now acceptable for publication, you may indicate that here to bypass the “Comments to the Author” section, enter your conflict of interest statement in the “Confidential to Editor” section, and submit your "Accept" recommendation.

Reviewer #1: (No Response)

Reviewer #2: All comments have been addressed

2. Is the manuscript technically sound, and do the data support the conclusions?

Reviewer #1: No

Reviewer #2: Yes

3. Has the statistical analysis been performed appropriately and rigorously? 

Reviewer #1: No

Reviewer #2: Yes

4. Have the authors made all data underlying the findings in their manuscript fully available?

Reviewer #1: Yes

Reviewer #2: Yes

5. Is the manuscript presented in an intelligible fashion and written in standard English?

Reviewer #1: No

Reviewer #2: Yes

6. Review Comments to the Author

Reviewer #1: I considered there is no significant improvement for our undestanding on the plastid phylogeny of Myristicaceae. One of the main findings of this study that Horsfieldia pandurifolia should be treated as Endocomia macrocoma subsp. prainii is not found in this study for the first time, but has been widely accepted in recent twenty years.

The authors did not follow my suggestions to combine more available data from Cai et al. (2021) and Li et al. (2021) for phylogenetic analyses.

Reviewer #2: The authors made a great progress on improving the manuscript, adequately addressing my comments. Yet the author could consider the following suggestions in further polishing their manuscript:

L58: "considered a high-quality raw material [3-6], implying a significant economic opportunity in"

L110-L113: I suggest to revise the sentence on colllection sites and specimens as "The collection sites included Nangunhe National Nature Reserve, Xishuangbanna National Nature Reserve and Yunnan Institute of Tropical Crops (YITC), Yunnan, China. All the live specimen was planting on site."

L139: "..., the accession numbers were ... "

L161: "..., DnaSP software ..."

L204: "and rpl32~trnL-UAG in the SSC. Among them, ..."

L259-260: "To determine the phylogenetic relationship of the 10 species of Myristicaceae, phylogenetic trees were reconstructed ..."

L270: The citation (http://www.plantplus.cn/rep/protlist) should be properly cited in Vancouver style!!!

L284: "regions, followed by coding regions, and most TRSs were located in ycf genes and non-coding"

L324-325: "... In this study, the phylogenetic relationships of 10 Myristicaceae species were inferred using ML and Bayesian methods."

Table 1 and elsewhere: please carefully check the accession numbers of your chloroplast genomes. The one of Knema cinerea (Poir.) Warb. should be "MN683756" but not "MN583756"!

7. PLOS authors have the option to publish the peer review history of their article (what does this mean?). If published, this will include your full peer review and any attached files.

Reviewer #1: No

Reviewer #2: **Yes: **Kwan Ho WONG

---

## [Author Response · Author response to Decision Letter 1]

3 Nov 2022

Dear Editors and Reviewers:

Thank you for your letter and for the reviewers’ comments concerning our manuscript entitled “Complete chloroplast genome sequence s of Myristicaceae species with the comparative chloroplast genomics and phylogenetic relationships among them” (ID: PONE-D-22-11654R1). Those comments are all valuable and very helpful for revising and improving our paper, as well as the important guiding significance to our researches. We have studied comments carefully and have made correction which we hope meet with approval. Revised portion are marked in red in the manuscript. Specific modifications have been replied to in detail in the "Response to Reviewer" file

---

## [Decision Letter · Decision Letter 2]

17 Jan 2023

Complete chloroplast genome sequences of Myristicaceae species with the comparative chloroplast genomics and phylogenetic relationships among them

PONE-D-22-11654R2

Dear Dr. mao,

We’re pleased to inform you that your manuscript has been judged scientifically suitable for publication and will be formally accepted for publication once it meets all outstanding technical requirements.

Kind regards,

Michael Scott Brewer, Ph.D.

Academic Editor

PLOS ONE

Additional Editor Comments (optional):

Reviewers' comments:

Reviewer's Responses to Questions

**Comments to the Author**

1. If the authors have adequately addressed your comments raised in a previous round of review and you feel that this manuscript is now acceptable for publication, you may indicate that here to bypass the “Comments to the Author” section, enter your conflict of interest statement in the “Confidential to Editor” section, and submit your "Accept" recommendation.

Reviewer #2: All comments have been addressed

2. Is the manuscript technically sound, and do the data support the conclusions?

Reviewer #2: Yes

3. Has the statistical analysis been performed appropriately and rigorously? 

Reviewer #2: Yes

4. Have the authors made all data underlying the findings in their manuscript fully available?

Reviewer #2: Yes

5. Is the manuscript presented in an intelligible fashion and written in standard English?

Reviewer #2: Yes

6. Review Comments to the Author

Reviewer #2: The authors made great efforts in improving the manuscript by addressing comments of mine and reviewer 1. I would suggest acceptance of this manuscript for publication.

7. PLOS authors have the option to publish the peer review history of their article (what does this mean?). If published, this will include your full peer review and any attached files.

Reviewer #2: **Yes: **Kwan Ho WONG

---

## [Editor Report · Acceptance letter]

9 Feb 2023

PONE-D-22-11654R2 

Complete chloroplast genome sequences of Myristicaceae species with the comparative chloroplast genomics and phylogenetic relationships among them 

Dear Dr. mao:

I'm pleased to inform you that your manuscript has been deemed suitable for publication in PLOS ONE. Congratulations! Your manuscript is now with our production department. 

Kind regards, 

on behalf of

Dr. Michael Scott Brewer 

Academic Editor

PLOS ONE